

# Estimates of the vitality and performances of *Apis mellifera mellifera* and hybrid honey bee colonies in Siberia: a 13-year study

Nadezhda V. Ostroverkhova[1,2], Svetlana A. Rosseykina[1],
Ilona A. Yaltonskaya[1] and Michail S. Filinov[1]

[1] Invertebrate Zoology Department, Biology Institute, Tomsk State University, Tomsk, Tomsk Region, Russia
[2] Department of Biology and Genetics, Siberian State Medical University, Tomsk, Tomsk Region, Russia

Corresponding author
Nadezhda V. Ostroverkhova,
nvostrov@mail.ru

## ABSTRACT

Honeybees display a great range of biological, behavioral, and economic traits, depending on their genetic origin and environmental factors. The high diversity of honeybees is the result of natural selection of specific phenotypes adapted to the local environment. Of particular interest is adaptation of local and non-local bee colonies to environmental conditions. To study the importance of genotype-environment interactions on the viability and productivity of local and non-local bee colonies, we analyzed the long-term dynamics of the main traits in dark forest bees (*Apis mellifera mellifera*) and hybrid colonies. From 2010 to 2022, a total of 64 colonies living in an apiary in Siberia, Russia, were monitored and tested to assess their biological, behavioral, and economic traits in a temperate continental climate. We detected significant correlations between the studied biological and behavioral indicators of the bee colony such as colony strength, overwintering ability, infection of colonies with diseases, hygienic behavior, and others. No relationships between the biological and economic (honey productivity) traits of bee colonies are shown. The overall result of our study is that local dark forest bee, *A.m.mellifera*, showed higher values for all analyzed traits than hybrid colonies. Compared to hybrids, dark forest bee colonies showed more gentleness, productivity, and survivorship. The results from our study indicate a specific local adaptation of the *A.m.mellifera* subspecies in a temperate continental climate. Siberia represents a unique region for the conservation of the dark forest bee. The creation of conservation areas is one way to protect local bee populations, well adapted to local environmental conditions, from uncontrolled importation of bee breeds from different regions.

## INTRODUCTION

The honey bee (*Apis mellifera* L.) is one of the most important managed insect pollinators worldwide (*Gallai et al., 2009*; *Danner et al., 2017*; *Hung et al., 2018*). In the last decade a

decrease in the number of honey bee colonies has been reported in many countries (*Potts et al., 2010a*; *van der Zee et al., 2014*; *Lee et al., 2015*; *EPILOBEE Consortium et al., 2016*; *Porrini et al., 2016*; *Calovi et al., 2021*). Bee declines have been attributed to many factors, such as pests and pathogens (*Higes, Martin & Meana, 2006*; *De Miranda & Genersch, 2010*; *Genersch et al., 2010*; *Guzmán-Novoa et al., 2010*; *Nazzi et al., 2012*; *Steinmann et al., 2015*), pesticides (*Frazier et al., 2008*; *Chauzat et al., 2009*; *vanEngelsdorp et al., 2009*; *Johnson et al., 2010*; *Di Noi et al., 2021*), habitat destruction (*Potts et al., 2010b*; *Zhang et al., 2022*), poor nutrition (*Alaux et al., 2010*; *Neumann & Carreck, 2010*; *Paudel et al., 2015*; *Steinhauer et al., 2018*), and combinations of stressors (*Nguyen et al., 2010*; *vanEngelsdorp & Meixner, 2010*; *Goulson et al., 2015*).

One important factor contributing to colony losses is the use of maladapted honeybees by beekeepers, such as bee colonies with low viability or bee strains that are not well adapted to local environmental conditions (*Costa et al., 2012*). Currently, large-scale introduction of commercial breeds has caused introgressive hybridization and the local extinction of native honey bee populations in many areas (*De la Rúa et al., 2009*; *Meixner et al., 2010*; *Potts et al., 2010a*; *Henriques et al., 2018*). For example, high queen production, lower defensive behaviour, active spring development, and high honey production led to now almost worldwide distribution of *Apis mellifera ligustica* and *Apis mellifera carnica* (*De la Rúa, Fuchs & Serrano, 2005*; *Uzunov et al., 2014*; *Bieńkowska, Łoś & Węgrzynowicz, 2020*; *Hoppe et al., 2020*). While the natural distribution of the Italian honeybee *A. m. ligustica* is the Italian Peninsula, and the "Carniolan bee" *A. m. carnica* was only distributed across central-eastern European countries (*De la Rúa et al., 2009*).

Import of non-local bee subspecies and the difficulty to control mating lead to uncontrolled gene flow between local and imported (commercial) bees within a geographic area. Introgressive hybridization modifies the genetic pool of local bee populations, leading to the loss of their genetic identity, low adaptability to environmental factors, and poor economic performance (*De la Rúa et al., 2009*; *Meixner et al., 2010*, *2013*; *Büchler et al., 2014*). The scale of these processes and their impact on the vitality of bee colonies is still unknown.

Non-local bee colonies and hybrid bees are poorly adapted to new habitat conditions, and strong colonies have suddenly become weak and died. Loss of adaptation to the local environment is thought to result in reduced colony survival, increased susceptibility to infections and other stressors (*Meixner et al., 2010*). It has been shown that locally adapted bee strains suffer less from elevated losses, are more viable and productive than non-local bees (*Costa, Lodesani & Bienefeld, 2012*; *Alqarni, Balhareth & Owayss, 2014*; *Francis et al., 2014*; *Hatjina et al., 2014*; *Al-Ghamdi et al., 2017*; *Taha & AL-Kahtani, 2019*; *Kovačić et al., 2020*).

Recently, evaluation of the viability and the performances of local and non-local bees has become an important research topic to investigate the vitality of colonies, survival rate, and the role of genotype–environment interactions in the health and performances of bee colonies (*Costa et al., 2012*; *Costa, Lodesani & Bienefeld, 2012*; *Büchler et al., 2014*; *Hatjina et al., 2014*; *Meixner et al., 2014*; *Meixner, Kryger & Costa, 2015*). The colony performances

are reported to be affected by internal (bee race, queen fecundity, colony size, stored food) and external factors (climate, management practices, nectar and pollen sources).

It should be noted that in Europe, research is carried out mainly on the most common subspecies, such as *A. m. carnica* (*Hoppe et al., 2020*; *Kovačić et al., 2020*) and *A. m. ligustica* (*Costa, Lodesani & Bienefeld, 2012*). Although in several regions of Europe, these two subspecies are now favored over the local honeybees such as *Apis mellifera mellifera* and *Apis mellifera siciliana*, conservation and reintroduction attempts have been initiated for some of these populations (*Jensen et al., 2005*; *Dall'Olio et al., 2007*; *Strange, Garnery & Sheppard, 2007*; *De la Rúa et al., 2009*). Studies evaluating the viability and productivity of these subspecies are rare (*Costa et al., 2012*; *Büchler et al., 2014*; *Hatjina et al., 2014*; *Guichard et al., 2020*), and now the dark forest bee *A. m. mellifera* is recognized as an endangered species by the European Society of Beekeepers (*Jensen & Pedersen, 2005*; *Soland-Reckeweg et al., 2009*; *Pinto et al., 2014*; *Muñoz et al., 2015*).

Of the 30 currently described subspecies of honeybees, ten subspecies are found in Europe. Based on morphological characteristics, European subspecies belong to three evolutionary lineages (M, C, and O) and are characterized by specific morphometric features (subspecies standard) (*Ruttner, 1988*; *De la Rúa et al., 2009*). In honeybees, classical morphometry involves comparing measurements of various body features such as body painting, size (proboscis, sternite, *etc.*), and parameters of the wing (angles, cubital index, hantel index, *etc.*). In addition to morphometry, the COI-COII intergenic region (between the cytochrome oxidase I and the cytochrome oxidase II) of the mtDNA, including P and Q(s) sequences, is powerful to characterize diversity between populations of the various evolutionary lineages and to determine the maternal origin of bee colonies (*Garnery, Cornuet & Solignac, 1992*). For example, subspecies of the lineage C (*A. m. carnica*, *A. m. ligustica*, *etc.*) have the shortest sequence of COI–COII mtDNA locus (variant Q); subspecies of lineage M (*A. m. mellifera*) are characterized by a longer sequence (one of the variants P2Q, P3Q, P4Q, or P5Q) (*Cornuet, Garnery & Solignac, 1991*; *Rortais et al., 2011*).

In Siberia, the honey bee was introduced at the end of the 18th century; it was a *A. m. mellifera* subspecies ("Dark forest bee"). The dark forest bee is the well adapted to the harsh climatic conditions of the northern region of Eurasia and has mastered the forest and forest steppe zones. In contrast to other honey bee subspecies, *A. m. mellifera* is characterized by a high level of adaptation to adverse environmental factors and greater resistance to some diseases. In Siberia, a dark forest bee is well adapted to the local climate and plant communities, but the wintering of bee colonies is controlled by humans (*Ostroverkhova et al., 2018*). In the second half of the 20th century and at the beginning of the current century, the sale and movement of commercial queens and packaged honeybees has occurred at large scale, and this process becomes widespread and almost uncontrollable.

The aim of our work was to comprehensively investigate the effects of genotype on the development, viability, and performance of bee colonies for a better understanding of the process of adaptation. We studied local subspecies (*A. m. mellifera*) and hybrids (result of

hybridization of the local *A. m. mellifera* subspecies and the introduced bee subspecies of southern origin) under environmental conditions of Siberia from 2010 to 2022.

## MATERIALS AND METHODS

### Research site

The research was carried out at the isolated apiary of the Tomsk State University, located in the Tomsk region (56°29′35″N, 85°10′26″E) (Fig. 1). The Tomsk region is in the geographic center of Siberia (southeast of the West Siberian Plain) and within the taiga zone. The climate is temperate continental; significant daily and annual amplitudes of temperature, and late spring and early autumn frosts; the average annual temperature is −0.6 °C. Winter lasts 5–6 months, summer is short and warm. The average temperature in January is −19.2 °C, in July– +18.1 °C. The frost-free period is 100–120 days. Precipitation 435 mm (*Ostroverkhova et al., 2018*).

Initially, *A. m. mellifera* bee colonies lived in the apiary. At the beginning of this century, bee colonies of the subspecies *A. m. carpathica* (a derivative of *A. m. carnica*) were brought to the apiary, which led to a "deterioration" of the genotypic composition of most colonies. In 2015, in Siberia, we found two surviving *A. m. mellifera* populations, Yenisei and Ob (*Ostroverkhova et al., 2018*; *Ostroverkhova, 2020*). The Yenisei population is localized in the inaccessible taiga of the Krasnoyarsk Territory. The Ob population is an isolated population in the north of the Tomsk region. Bee colonies from Siberian populations became the basis for the formation of a breeding core of the *A. m. mellifera* bee farm. Since 2016, most colonies in the apiary are purebred dark forest bee colonies. According to the long-term dynamics of the analyzed traits of colonies, the most productive colonies were selected, and the least productive ones were culled.

### Study design

A comprehensive assessment of bee colonies, including biological, behavioral, and economic characteristics, as well as the impact of genetic factor on their development, was carried out at the apiary from 2010 to 2022. The same 64 bee colonies were studied annually.

At the start of colony evaluations in April 2010, we studied the subspecies composition of honeybees in the apiary. Every year, we monitored the bee colonies for their compliance with the *A. m. mellifera* standard (Subspecies discrimination). In 2015–2016, in colonies that do not meet the *A. m. mellifera* standard, a queen was changed to a purebred one.

The long-term dynamics of the main characteristics of bee colonies in the period from 2010 to 2022 has been studied. In addition, we separated two periods: (1) the period from 2010 to 2015, when the apiary had a heterogeneous composition of bee colonies (mainly hybrids); (2) the period from 2016 to 2022, when purebred dark forest bees (*A. m. mellifera*) prevailed in the apiary.

The colonies were managed according to a common test protocol and uniform methods, which included colony inspections at regular intervals, and a continuous assessment of their health status. The inspection of bee colonies included: spring census after wintering (time range between April to May); summer census before the end of honey harvest

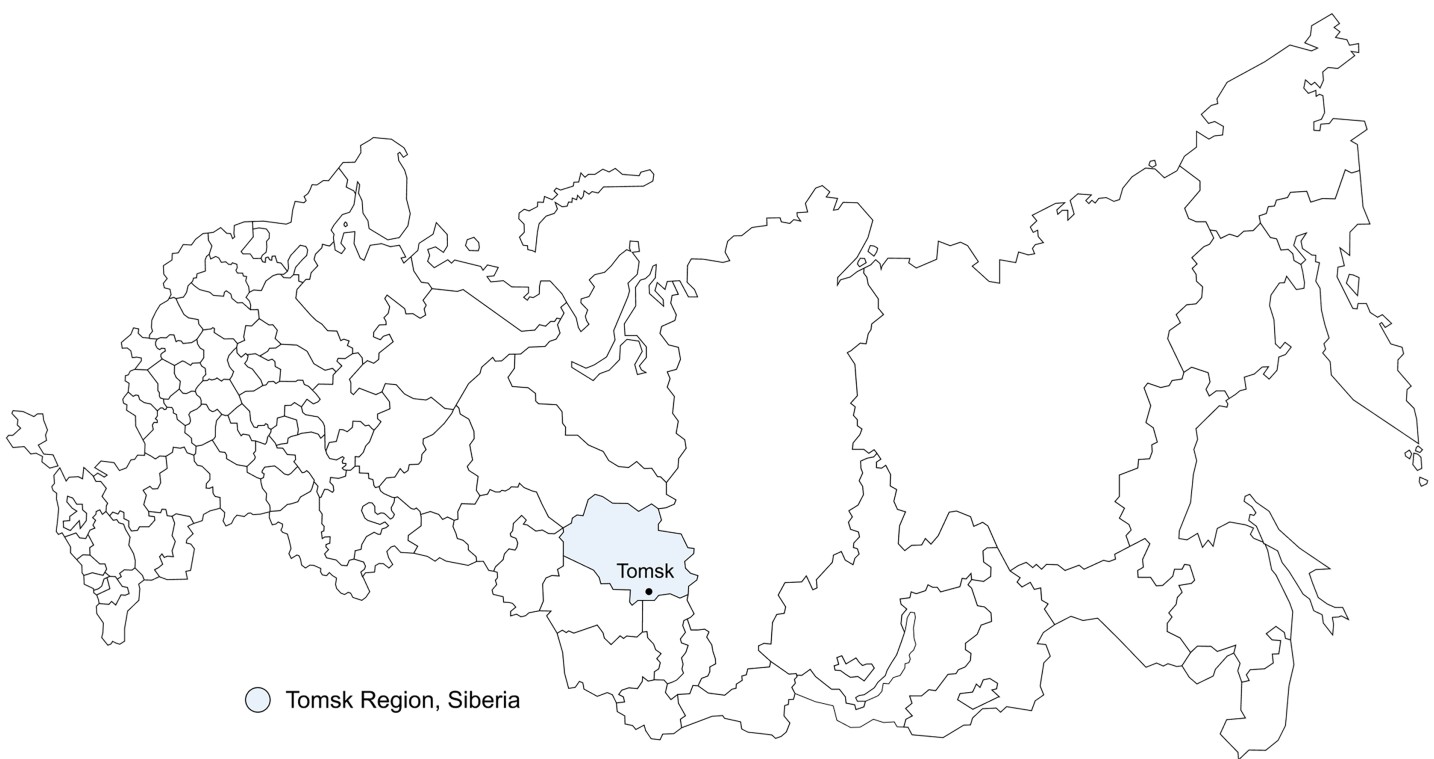

**Figure 1 Localization map of the Tomsk Region, Russia, and the apiary where the research was carried out (in the vicinity of the city of Tomsk).**

(between mid-July and mid-August), and autumn census performed at end of the season (from September to October).

Bee colonies were assessed for the traditional selection traits such as honey productivity, gentleness, swarming tendency, as well as for traits associated with colony vitality, such as colony development, overwintering ability, disease resistance. We used colony assessment methods described by *Costa et al. (2012)*, *Büchler et al. (2014)*, and *Brandorf & Ivoilova (2015)*. We divided all tested traits into biological (colony strength, overwintering ability, disease resistance), behavioral (gentleness, hygienic behavior, swarming tendency) and economic (honey production).

At the last stage of the study, we conducted a comparative analysis of the viability and productivity of purebred (*A. m. mellifera*) and hybrid colonies from 2012 to 2014. In 10 bee colonies (five purebred and five hybrid colonies), we assessed some biological (queen egg laying, spring development of colonies), behavioral (gentleness, swarming tendency, calmness on the comb, hygienic behavior, cleanliness of the beehive), and economic (honey productivity) traits.

### Assessment of the origin of the bee colony

*Subspecies discrimination.* About 25–30 young bees per colony were collected from the brood-nest area and preserved in 96% ethanol for subspecies identification. Then these bees were analyzed by the following subspecies discrimination methods: mitochondrial DNA (mtDNA) and morphometric analyses. MtDNA analysis (variability of the locus

*COI–COII*) was used to determine the origin of the bee colony on the maternal line. DNA isolation and polymerase chain reaction (PCR) was carried out according to *Nikonorov et al. (1998)*.

The samples of bees with variants PQQ or PQQQ of the *COI–COII* mtDNA locus were submitted to morphometric analysis to confirm that the bees belonged to the *A. m. mellifera* subspecies (evolutionary lineage M). Morphometric measurements of the collected samples included the following main wing parameters (wing venation): the cubital index, the hantel index, and the discoidal shift (*Cauia et al., 2008*). We also examined the body painting of the bees. If the morphometric parameters of bees correspond to the dark forest bee's standard, this colony was considered potentially "pure" *A. m. mellifera*. If the morphometric parameters of bees were not consistent with *A. m. mellifera* breed standards, we considered bee colonies as hybrids.

## Assessment of biological, behavioral, and economic parameters of bee colonies

*Colony strength*: estimated by the number of adult bees in spring (after wintering) and in autumn (before wintering). For each colony, the number of frames covered by bees was recorded, and the frames were converted to appropriate weight units (one frame corresponds to 250 g of bees). For example, a strong colony weighs 2.0–2.5 kg (8–10 frames) (*Brandorf & Ivoilova, 2015*).

*Spring development of colony*: assessed by changes in colony strength from April (after wintering) to July–August (before honey collection). Colonies were assessed every week.

*Overwintering ability*: determined as a percentage of bees that died during wintering, according to the formula:

$$\frac{(\text{Colony strength in the previous autumn} - \text{Colony strength in spring})(\text{frames})}{\text{Colony strength in the previous autumn (frames)}} \times 100$$

*Queen egg laying* (average eggs laid per day): calculated by indirect counting of sealed brood every 12 days. The area of sealed brood was used to determine the number of bees that hatch in the next 12 days after counting (*Brandorf & Ivoilova, 2015*).

*Disease resistance*: determined by the level of infection of bee colonies with major diseases (varroosis, nosemosis, and Chalkbrood disease). The infection of bee colonies was determined in the spring during the revision of colonies after wintering. Bees were collected from outer frames.

*Varroa infestation*. Sampling bees to determine *V. destructor* infestation levels was performed in spring. The bees (about 30 g) were floated in soapy water for about 30 min and shaken for one min, to dislocate the mites from the bees (*Costa et al., 2012*). Bees and mites were then separated by straining, and the mites were counted. Data were expressed as number of mites per 100 bees. Three degrees of *Varroa* infestation were distinguished: weak–up to two mites per 100 bees, medium–up to four mites per 100 bees, and strong–over 4 mites per 100 bees.

*Nosema infection* was diagnosed by clinical signs of bees, such as increase in the abdomen, diarrhea, trembling of the wings and death of workers, as well as contamination

of the walls and frames of the beehives, which is typical for type A nosemosis (causative agent *Nosema apis*).

In the spring of 2017, a random sample of 10 colonies without clinical signs of nosemosis was examined. From each test colony 50–60 forager bees were immediately frozen for discrimination of *Nosema* species (*N. apis* and/or *N. ceranae*). DNA was extracted from the midgut of bees (a pool of bees was formed) using a DNA purification kit, PureLink™ Mini (Invitrogen, Carlsbad, CA, USA) according to the manufacturer's protocol. Then the samples were submitted to duplex-PCR (*Martín-Hernández et al., 2007*).

*Ascosphaera apis infection (Chalkbrood disease).* Chalkbrood disease can be easily diagnosed using visual detection methods. Hives infected by Chalkbrood disease symptom appeared to have hard, shrunken chalk-like mummies in the brood and surrounding the entrance to the hive (*Flores, Gutiérrez & Espejo, 2005*; *Aronstein & Murray, 2010*; *Kim et al., 2023*). *Ascosphaera apis* infection of colonies was assessed by analyzing infected larvae per comb. There are three degrees of infection depending on the number of dead larvae per comb: weak–up to 10 dead larvae, medium–from 11 to 100, strong–more than 100 dead larvae (*Domatsky & Domatskaya, 2022*).

To assess the diseases resistance of bees, the indicators "hygienic behavior" and "cleanliness of the beehive" were also used. *Hygienic behaviour* of the colony was assessed by the ability of the bees to open cells and remove diseased larvae by adult bees (sanitizing ability) using the five-point system: 1–no sanitizing ability; 2–unsatisfactory ability; 3–satisfactory ability; 4–good ability; 5–exceptional sanitizing ability. *Cleanliness of the beehive* was judged after wintering using the five-point system: 1–very heavy fecal contamination of the hive; 2–severe fecal contamination of the hive; 3–average contamination; 4–slight contamination; 5–clean beehive.

We assessed the following behavioral characteristics by the four-point system according to *Costa et al. (2012)*: gentleness, swarming tendency, and calmness on the comb during inspection.

*Honey production.* Honey yield was taken as weight difference of combs before and after extracting the honey (*Ruttner, 1972*).

## Statistical analysis

Systematization of the source data, statistical processing, and statistical analysis of the results were accomplished using the computer software Microsoft Office Excel 2016. Descriptive statistics were applied to organize the data. The data was expressed in mean ±standard error of the mean (M ± m). A Student's t-test was used to compare the mean values for independent variables. Correlation analysis (Pearson's correlation coefficients, $r$ and value/coefficient of determination, $R^2$ value) was performed to determine whether there was a correlation between biological, behavioral, and economic traits of bee colonies. Fisher's test at $p \leq 0.05$ were used to determine the significance of the relationships between indicators of bee colonies. In general, the level of significance was expressed as *$p < 0.05$, **$p < 0.01$, and ***$p < 0.001$.

For comparative analysis of biological traits (queen egg laying, spring development of colonies) between purebred (*A. m. mellifera*) and hybrid colonies in 2012–2014, R statistics (Version 4.3.2) were also used. We used the Durga R package (DurgaDiff and Durga Plot functions). The Durga Diff function estimates between-group effect sizes. Standardised effect sizes were calculate using Cohen's *d*, standardized mean difference. A *d* of 0.2 or smaller is considered to be a small effect size, a *d* of around 0.5 is considered to be a medium effect size, and a *d* of 0.8 or larger is considered to be a large effect size. The Durga Plot function is used to visualize the results (*Khan & McLean, 2023*).

## RESULTS

To determine the origin of bee colony on the maternal line, we studied the variability of the locus *COI–COII*. We established that 75.5% of bee colonies on the maternal line originate from the dark forest bee (PQQ variant identified), and 24.5% of colonies–from C-lineage subspecies (Q variant identified). Morphometric analysis confirmed that the majority of the studied bee colonies could be classified as hybrids. For example, Table 1 shows the morphometric indicators of some purebred (*A. m. mellifera*) and hybrid bee colonies.

### Long-term dynamics of biological indicators of bee colonies (2010–2022)

In the climatic conditions of Siberia, the strength of a bee colony in spring should be at least 2 kg. The mean value of the colony strength changed from 1.90 kg (2010) to 2.75 kg (2019) in spring and from 2.33 kg (2022) to 3.18 kg (2016) in autumn (Fig. 2). Statistically significant differences were shown between the average colony strength in spring and in autumn ($t_s = 3.46$, $p = 0.002$).

Since 2015, purebred queens have been introduced into most hybrid colonies, and a significant increase in colony strength has been shown (Fig. 2). Since 2019, due to changes in beekeeping technology (breeding apiary), namely the active breeding queens, lower colony strength values have been noted. In this regard, the colony strength is decreasing from 2016 to 2022.

Overwintering ability of bee colonies, assessed by "death of bees after wintering" indicator is characterized by a gradual increase (Fig. 2). From 2010 to 2022 the mean value of the indicator "death of bees" decreased from 18% in 2010 to 12% in 2019–2022.

Correlations between the colony strength in spring and the colony strength in autumn, as well as the colony strength in spring and death of bees after wintering were very significant. For the entire period of research (2010–2022), a strong association was detected between the colony strength in spring and the colony strength in autumn ($r = 0.706$, $p < 0.01$), the colony strength in spring and death of bees ($r = -0.754$, $p < 0.01$), and death of bees decreases as the colony strength increases. For the first period of the study, when hybrid colonies prevailed in the apiary (2010–2015), a significant correlation was showed only between colony strength in spring and death of bees after wintering ($r = -0.918$, $p < 0.01$). No correlations were found between the strength of the colony in spring and autumn ($r = 0.684$, $p > 0.05$). On the contrary, for the second period of research, when purebred colonies dominated in the apiary (2016–2022), a high correlation was

**Table 1 Morphometric parameters (wing venation) of honeybee workers from purebred (*A.m.mellifera*) and hybrid colonies.**

| Bee colony, № | Cubital index, standard units | | Hantel index, standard units | | Discoidal shift, % | | |
|---|---|---|---|---|---|---|---|
| | Lim | M ± m | Lim | M ± m | – | 0 | + |
| 1 | 1.17–1.81 | 1.51 ± 0.03 | 0.693–0.923 | 0.808 ± 0.010 | 100.0 | 0 | 0 |
| 2 | 1.15–1.89 | 1.52 ± 0.04 | 0.779–0.919 | 0.849 ± 0.011 | 100.0 | 0 | 0 |
| 3 | 1.20–1.67 | 1.45 ± 0.02 | 0.723–0.900 | 0.837 ± 0.009 | 97.0 | 3.0 | 0 |
| 4 | 1.32–2.00 | 1.57 ± 0.04 | 0.712–0.851 | 0.789 ± 0.010 | 100.0 | 0 | 0 |
| 5 | 1.60–2.60 | 2.04 ± 0.04 | 0.753–1.000 | 0.875 ± 0.009 | 71.1 | 24.4 | 4.5 |
| 6 | 1.50–2.20 | 1.88 ± 0.03 | 0.756–1.000 | 0.868 ± 0.065 | 78.1 | 21.9 | 0 |
| 7 | 1.30–2.80 | 1.77 ± 0.03 | 0.741–0.982 | 0.847 ± 0.010 | 22.8 | 62.9 | 14.3 |
| Standard for *Apis mellifera mellifera* | | | | | | | |
| I | 1.3–2.1 | 1.7 | 0.600–0.923 | No data | No data | | |
| II | 1.3–1.9 | 1.5–1.7 | 0.600–0.923 | No data | 91–100 | 5–10 | 0 |

**Notes:**
Bee colonies 1, 2, 3, and 4–dark forest bee *A.m.mellifera*, colonies 5, 6, and 7–hybrids. Minimum 30 samples from each bee colony were studied. Lim, limits of value of the sing; M ± m, average value of the sign ± the standard error of the mean. I, European breed standard based on values of cubital and hantel indexes (*Cauia et al., 2008*). II, Russian breed standard.

shown between the colony strength in spring and the colony strength in autumn ($r = 0.821$, $p < 0.05$), but no correlations were found between the strength of the colony in spring and death of bees after wintering ($r = 0.348$, $p > 0.05$). Correlation is not significant between the colony strength in autumn and death of bees after wintering.

## Disease infestation of bee colonies

During the test period, varroosis (caused by the *Varroa destructor* mite), nosemosis (caused by microsporidia of the *Nosema* genus), and Chalkbrood disease (caused by the *Ascosphaera apis* fungus) were identified in bee colonies (Fig. 3). There is a gradual decrease in the infection of colonies in general and for individual diseases (except for 2013 and 2014). The number of infected colonies decreased from 37.5% in 2010 to 0% in 2020.

Initially, in 2010, nosemosis and varroosis were most common in the apiary (25.0% and 20.3% of bee colonies, respectively). There was an increase of colonies with Chalkbrood disease (up to 10.9%) in 2011–2012, and with varroosis (up to 17.2%) in 2013–2014. A significant proportion of colonies were infected with two pathogens: *Varroa* and *Nosema* (10.5–27.8% of infected colonies); *Nosema* and *Ascosphaera apis* (4.2–5.6%); *Varroa* and *Ascosphaera apis* (5.6–25.0%). Interestingly, all hybrid colonies were infected with *V. destructor* and/or *Ascosphaera apis*, while only 25% of purebred colonies were infected with varroosis.

It should be noted that for nosemosis, the results based on clinical signs of bees examined in spring after overwintering are presented (Fig. 3). We analyzed bee abdomen size, diarrhea, wing trembling, and death of individuals, as well as contamination of the walls and frames of the hive with bee excrements, which is characteristic of type A nosemosis caused by *Nosema apis*.

In 2017, a study of ten randomly selected colonies without clinical signs of nosemosis showed the presence of microsporidia spores. Using the PCR, both pathogens (*N.apis* and

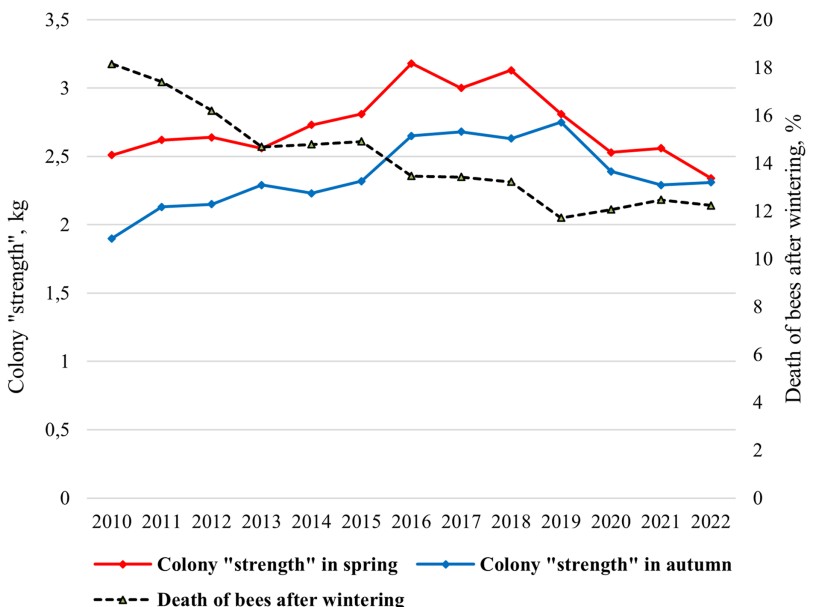

**Figure 2 Long-term dynamics of biological indicators "colony strength" and "death of bees after wintering" of bee colonies in the period from 2010 to 2022.**

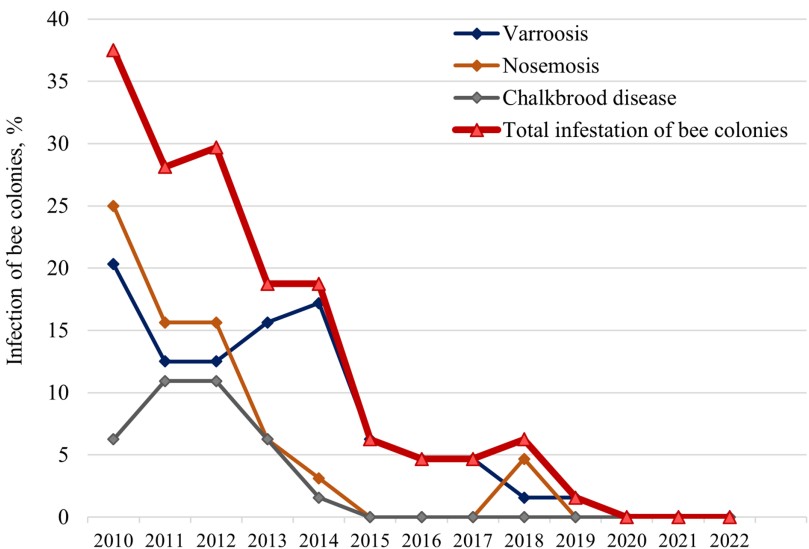

**Figure 3 Long-term dynamics of infection of bee colonies with major diseases (varroosis, nosemosis, and Chalkbrood disease).** The study was conducted in early spring after wintering annually from 2010 to 2022.

*N. ceranae*) were detected in all colonies, and *N. apis* prevailed in most colonies. Thus, both *Nosema* species was detected in honeybees, but the bee colonies did not get sick and were resistant to nosemosis.

We also investigated the correlations between the infestation of bee colonies and their biological traits from 2010 to 2015 (the test period when the main diseases of bee colonies were detected) (Table 2). Statistically significant correlations were found between the total

**Table 2 Correlation analysis between the infestation of bee colonies and biological traits of colonies from 2010 to 2015.**

Coefficient of determination, $R^2$

| Infection of bee colonies–Colony strength in autumn | | | | Infection of bee colonies–Colony strength in spring | | | | Infection of bee colonies–Death of bees | | | |
|---|---|---|---|---|---|---|---|---|---|---|---|
| Total | *Varroa* | *A.apis* | *Nosema* | Total | *Varroa* | *A.apis* | *Nosema* | Total | *Varroa* | *A.apis* | *Nosema* |
| 0.66* | 0.54 | 0.39 | 0.63 | 0.82* | 0.42 | 0.19 | 0.89** | 0.72* | 0.14 | 0.35 | 0.87** |

Notes:
The following designations were used: Total – infection of bee colonies with all diseases (varroosis, nosemosis, Chalkbrood disease); *Varroa* – *Varroa destructor* (varroosis); *A. apis* – *Ascosphaera apis* (Chalkbrood disease); *Nosema* – *Nosema apis* (type A nosemosis). A significant correlation is marked with star, and *, ** correspond to $p < 0.05$ and $p < 0.01$, respectively.

infestation of bee colonies and their spring strength ($R^2 = 0.82$, F = 18.22, $\alpha = 0.05$), autumn strength ($R^2 = 0.66$, F = 7.76, $\alpha = 0.05$), and death of bees ($R^2 = 0.72$, F = 10.29, $\alpha = 0.05$).

A high correlation was also shown between the infection of bee colonies with nosemosis and the colony strength in spring ($R^2 = 0.89$, F = 32.36, $\alpha = 0.01$), as well as the death of bees ($R^2 = 0.87$, F = 26.77, $\alpha = 0.01$) (Table 2). No correlations were found between the biological traits of colonies and the infection of colonies with varroosis and Chalkbrood disease ($p > 0.05$).

We also assessed parameter "cleanliness of the beehive after wintering", as indicator of the resistance of colonies to diseases. During the test period (2010–2022), in colonies, the value of sign "cleanliness of the beehive" changed from three points (average pollution of the beehive, *i.e.*, several dozen spots of excrements) to five points (clean beehive). There is a gradual increase of the mean value of sign from 3.833 points (2010) to 4.969 points (2021). A high, statistically significant negative correlation ($R^2 = 0.96$, F = 264, $\alpha = 0.01$) was shown between "cleanliness of the beehive" and "death of bees" signs of bee colonies during the entire test period (2010–2022). A correlation was found between these indicators both for hybrid bees (2010–2015) – $R^2 = 0.93$ (F = 53.14, $\alpha = 0.01$), and for purebred bees (2016–2022) – $R^2 = 0.68$ (F = 10.63, $\alpha = 0.05$).

## Long-term dynamics of economic indicators of bee colonies (2010–2022)

For the final assessment of bee colonies, we analyzed the average honey productivity of the colonies in the apiary from 2010 to 2022 and compared it with the honey productivity of the control bee colony. A strong, randomly selected bee colony was used as a control.

The average honey productivity of bee colonies changed during the entire study period from 32.4 kg in 2010 to 65.3 kg in 2019 and 2020 (Fig. 4). Starting from 2016, a significant increase in the honey productivity of colonies in the apiary has been shown. Statistically significant differences ($t_s = 4.75$, $p < 0.001$) between the mean values of honey productivity of hybrid colonies (2010–2015) and purebred colonies (2016–2022) were revealed.

Interestingly, from 2017 to 2022, the average honey productivity of colonies changed slightly (from 61.0 to 65.3 kg), compared with the period from 2010 to 2015 (from 32.4 to 57.4 kg). For example, in hybrid colonies, in a favorable year 2012 (the average temperature for the period May–July was more than 17 °C), a high value of honey

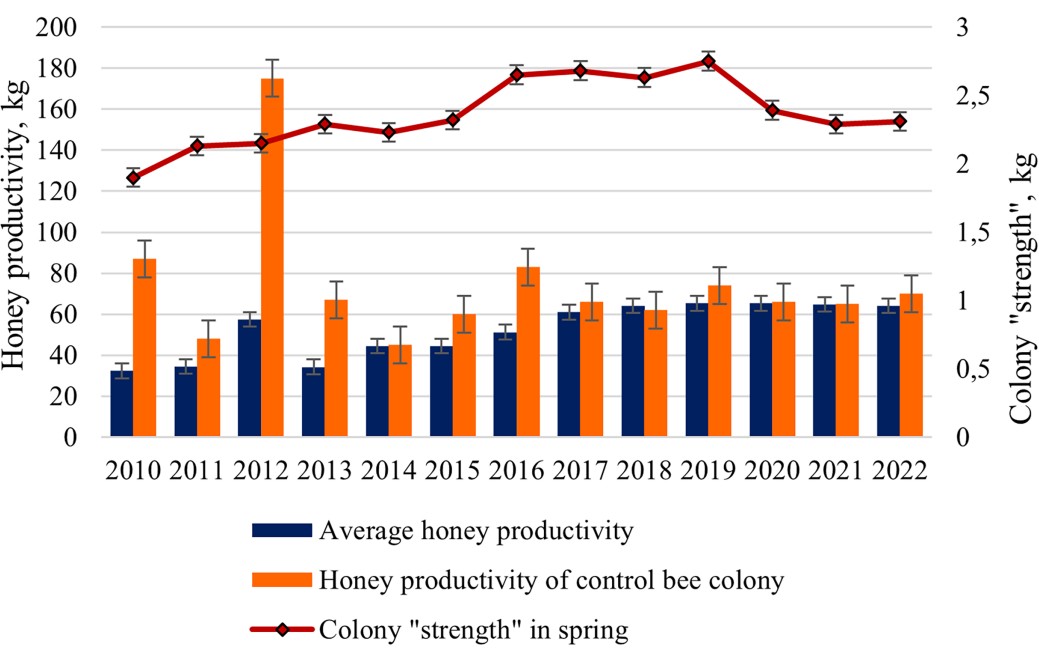

**Figure 4 Long-term dynamics of honey productivity (average and control colony) of bee colonies and the colony "strength" in the spring from 2010 to 2022.**

productivity (57.4 kg) was observed. In less favorable seasons, such as 2010 and 2013 (the average temperature for the period May-July was only 13 °C), honey productivity was 32.4 and 34.2 kg, respectively. In purebred colonies, in a favorable year 2020 (the average temperature for the period May–July was more than 16 °C) and in less favorable year 2018 (the average temperature for the period May–July was only 13 °C), honey productivity was the same (64–65 kg).

For the periods compared, there are also differences in the average honey productivity of test colonies and honey productivity of the control colony. For 2010–2015 (the predominance of hybrid colonies in the apiary), higher values of honey productivity for the control bee colony compared to the average honey productivity of the colonies are shown. For example, in 2010, these values were 87 and 32.4 kg, respectively. For 2017–2022 (the predominance of purebred colonies in the apiary), close values of the average honey productivity (61–65 kg) and the honey productivity of the control colony (62–74 kg) were noted. Of particular interest is the year 2012, characterized by favorable weather conditions: the super honey productivity in the control bee colony (175 kg) was shown; average honey productivity was only 57.4 kg.

### Comparative study on the vitality and performances of *Apis mellifera mellifera* and hybrid honeybee colonies (2012–2014)

To assess the significance of the genetic component in the development of colonies, a comparative analysis of some biological, behavioral, and economic traits in purebred *A. m. mellifera* and hybrid colonies was conducted from 2012 to 2014.
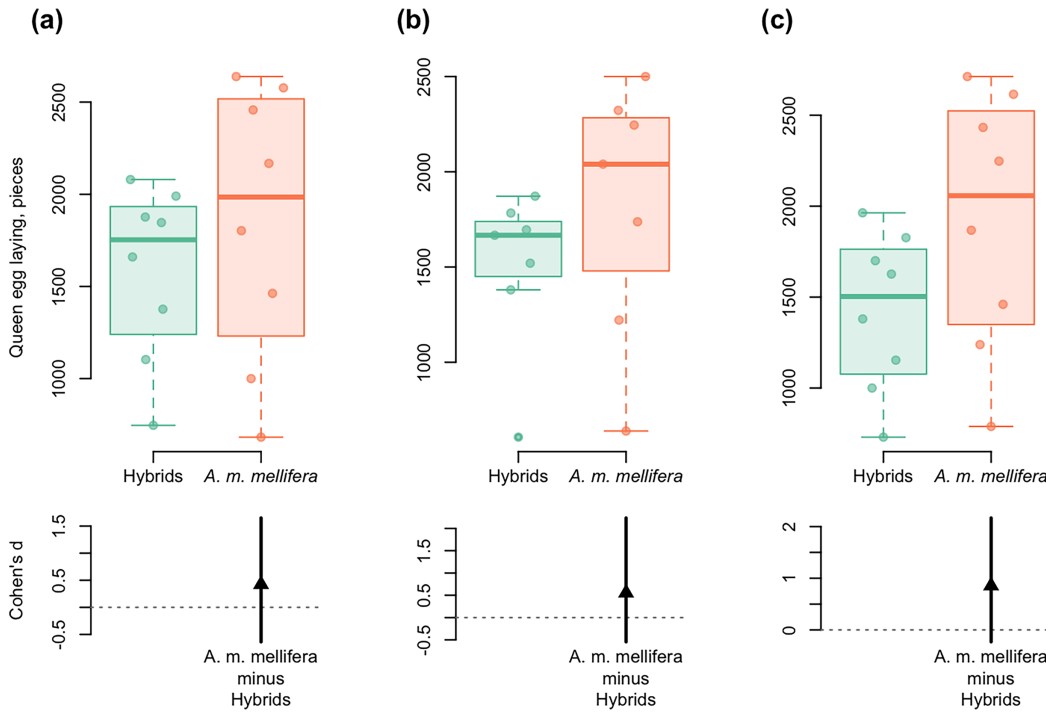

**Figure 5 DurgaPlots showing differences in queen egg laying between *A. m. mellifera* and hybrid bees in different years.** (A) 2012–*A.m.mellifera* ($n$ = 8, M = 1,848.594, SD = 742.51); hybrids ($n$ = 8, M = 1,585.001, SD = 470.02); (Cohen's $d$: 0.42, 95%, CI [−0.64 to 1.65]); (B) 2013–*A.m.mellifera* ($n$ = 7, M = 1,815.00, SD = 671.37); hybrids ($n$ = 7, M = 1,503.33, SD = 427.59); (Cohen's $d$: 0.55, 95%, CI [−0.55 to 2.23]); (C) 2014–*A.m.mellifera* ($n$ = 8, M = 1,920.34, SD = 700.96); hybrids ($n$ = 8, M = 1,422.50, SD = 432.08); (Cohen's $d$: 0.86, 95%, CI [−0.23 to 2.17]). Note: n, the number of measurements; M, mean; SD, standard deviation; CI, confidence interval. Here and in Fig. 6: Left axis represents group data and bottom region represents effect size statistics. Horizontal lines are drawn from the means of each group. Boxplots exhibit group data, the arrow bellow exhibits the distribution of bootstrapped Cohen's $d$ (standardized mean difference), the arrowhead represents Cohen's $d$, and vertical bar shows 95% confidence interval of Cohen's $d$. The boxplots display the group median and the 75th and 25th percentiles. The whiskers extend to the minimum and maximum values but exclude outliers that are beyond 1.5 times the interquartile range. Circles near to boxes indicate individual values.

In all the years studied, a similar dynamic of "average eggs laid per day" was shown for purebred and hybrid colonies. At the beginning of colony development, the queen egg laying was the same, then it was significantly higher in *A. m. mellifera* than in hybrids. Cohen's standardized mean difference ranged from $d$ = 0.42 in 2012 to $d$ = 0.86 in 2014 (95% CI) (Fig. 5). The average values of the indicator "average eggs laid per day" differed significantly between hybrid and purebred *A. m. mellifera* colonies in all the years studied ($t_s$ > 5.8, $p$ < 0.001).

Spring development of *A. m. mellifera* colonies was stronger and more active than in hybrids. Cohen's standardized mean differences were as follows: $d$ = 0.40 in 2012; $d$ = 0.30 in 2013; $d$ = 0.43 in 2014 (95% CI) (Fig. 6).

The *A. m. mellifera* bees were more gentleness than the hybrids. Purebred bees had a slight or controlled tendency to swarm, while hybrids often came to the swarming state. The dark forest bees showed exceptional or good sanitizing ability, and the beehives were

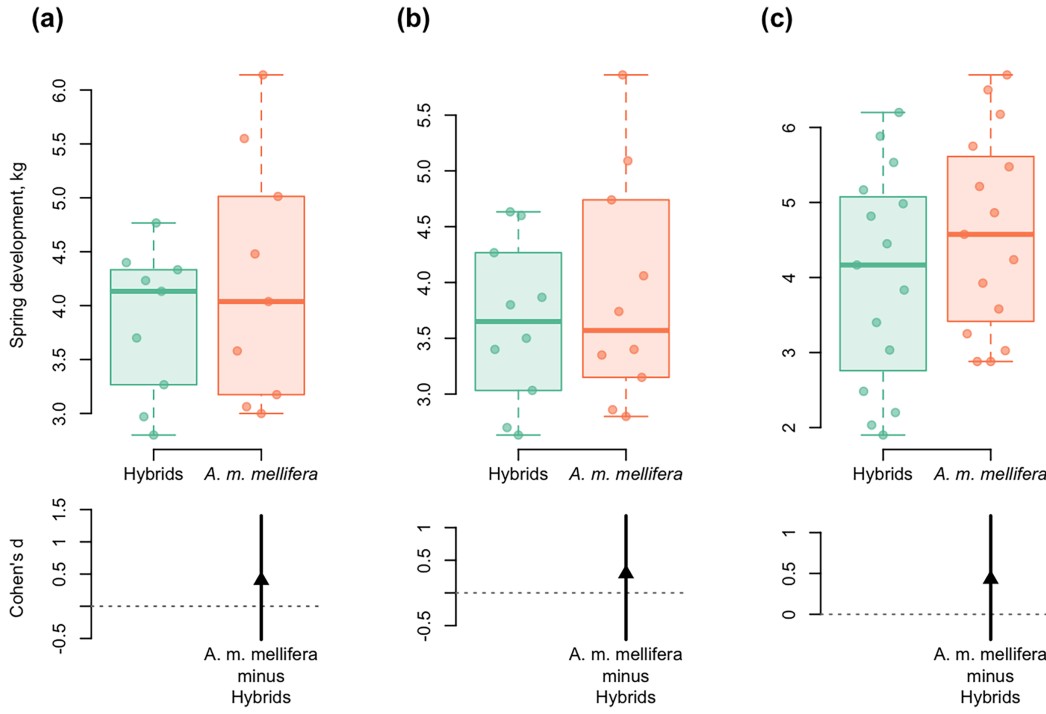

**Figure 6 DurgaPlots showing differences in spring development between purebred *A. m. mellifera* and hybrid colonies in different years.** (A) 2012–*A.m.mellifera* (*n* = 9, M = 4.23, SD = 1.15); hybrids (*n* = 9, M = 3.85, SD = 0.69); ( Cohen's *d*: 0.40, 95%, CI [−0.52 to 1.40]); (B) 2013–*A.m.mellifera* (*n* = 10, M = 3.90, SD = 1.02); hybrids (*n* = 10, M = 3.64, SD = 0.72); (Cohen's *d*: 0.30, 95%, CI [−0.71 to 1.18]); (C) 2014–*A.m.mellifera* (*n* = 15, M = 4.60, SD = 1.33); hybrids (*n* = 15, M = 4.00, SD = 1.44); (Cohen's *d*: 0.43, 95%, CI [−0.31 to 1.21]). Note: *n*, the number of measurements; M, mean; SD, standard deviation; CI, confidence interval.

mostly clean. In hybrid colonies, the sanitizing ability was lower (often good), and the hives had slight fecal contamination. Indicators of the resistance of bees to diseases such as "hygienic behavior" and "cleanliness of the beehive after wintering" differs between dark forest bees and hybrids (Table 3).

Honey productivity was the final assessment of purebred *A. m. mellifera* colonies and hybrid bees. In purebred colonies, the average honey productivity was 71.75 kg in 2012, 36.5 kg in 2013 and 38.0 kg in 2014. In hybrid colonies, the average honey productivity was 64.67 kg in 2012, 21.5 kg in 2013 and 33.0 kg in 2014. Purebred *A. m. mellifera* and hybrid colonies differed statistically significantly in honey productivity only in 2013, which was characterized by unfavorable weather conditions ($t_s$ = 3.24, *p* = 0.048).

In Table 4, a comparative description of the studied biological, behavioral, and economic traits of purebred *A. m. mellifera* and hybrid colonies are presented (Table 4).

## DISCUSSION

This study presents the first comprehensive analysis of the honeybees in Siberia, including an assessment of the biological, behavioral, and economic traits of bee colonies. The long-term dynamics of the main traits of bee colonies revealed the correlations between biological and behavioral characteristics, such as colony strength, death of bees after

**Table 3 Average values of behavioral traits for dark forest bees and hybrids during three seasons (2012–2014).**

| Behavioral trait | Dark forest bee (five colonies) | | | | Hybrid colonies (five colonies) | | | |
|---|---|---|---|---|---|---|---|---|
| | 2012 | 2013 | 2014 | M ± m | 2012 | 2013 | 2014 | M ± m |
| Gentleness[*] | 3 | 2 | 3 | 2.67 ± 0.41 | 3 | 2 | 2 | 2.33 ± 0.41 |
| Calmness on the comb during inspection[#] | 2.25 | 2 | 2 | 2.08 ± 0.10 | 3 | 2.33 | 2.67 | 2.67 ± 0.24 |
| Swarming tendency[##] | 2.25 | 2.75 | 2.25 | 2.42 ± 0.20 | 1.33 | 1.67 | 1.33 | 1.44 ± 0.14 |
| Hygienic behavior[&] | 5 | 4.75 | 4.50 | 4.75 ± 0.18 | 4.33 | 4 | 4 | 4.11 ± 0.13 |
| Cleanliness of the beehive[&&] | 5 | 4.75 | 4.75 | 4.83 ± 0.10 | 3.67 | 4 | 4 | 3.89 ± 0.13 |

Notes:
M ± m, average value of the sign ± the standard error of the mean.
[*] – Behavior of bees when inspecting the colony: 2 points–calm; 3 points–restless (Costa et al., 2012).
[#] – Calmness on the comb: 2 points–Bees partly leave their combs and cluster in the edges of frames and supers; 3 points–Bees are moving, but do not leave their combs during treatment (Costa et al., 2012).
[##] – Swarming tendency: 1 point–test colony swarmed; 2 points–swarming was difficult to control; 3 points–controlled swarming; 4 points–the colony has shown no swarming tendency (Costa et al., 2012).
[&] – Hygienic behavior: 3 points–satisfactory sanitizing ability; 4 points–good sanitizing ability; 5 points–exceptional sanitizing ability.
[&&] – Cleanliness of the beehive: 3 points–average fecal contamination of the hive; 4 points–slight fecal contamination of the hive; 5 points–clean beehive.

**Table 4 Comparative characteristics of biological, behavioral, and economic traits in purebred *A.m.mellifera* and hybrid colonies from 2012 to 2014.**

| Trait[*] | A. m. mellifera | Hybrid colonies | Year[**] |
|---|---|---|---|
| **Biological** | | | |
| Colony strength, spring, kg | 2.0–3.7 | 1.8–3.2 | 2014 |
| Colony strength, autumn, kg | 2.5–4.4 | 1.8–4.0 | 2013 |
| Overwintering ability, % | 12–15 | 14–21 | |
| Spring development, kg | 6.1–6.7 | 4.7–6.3 | 2012–2014 |
| Queen egg laying, pieces | 2,350–2,650 | ≤2,000 | 2012–2014 |
| **Behavioral** | | | |
| Gentleness | Calm | Restless | |
| Swarming tendency | Low degree | High degree | |
| Infestation with major diseases | Chalkbrood disease (25% colonies) | Varroosis, Chalkbrood disease (100% colonies) | |
| Cleanliness of the beehive, points | 3–5 | 3–5 | |
| **Economic** | | | |
| Honey productivity, kg | 49–109 | 50–96 | 2013 |

Notes:
[*] The minimum and maximum values of the traits are shown.
[**] The years are indicated when statistically significant differences were identified between the traits of purebred *A. m. mellifera* and hybrid colonies.

wintering, infection of colonies with diseases, and hygienic behavior. Interestingly, most of the correlations between biological and behavioral traits were found for hybrid colonies (the test period was from 2010 to 2015, when hybrid colonies predominated in the apiary). No correlation was found between biological and economic indicators. For example, no relationship between honey productivity and main biological traits, such as colony strength in spring ($R^2 < 0.20$, $p > 0.05$), was found.

*The colony strength* is one of the main indicators of the bee colony, determining its growth, behavior, and survival, including ability to survive over the winter and to reproduce by swarming (*Harris, 2009*; *Delaplane, Van der Steen & Guzman, 2013*).

The colony strength is greatly influenced by geographical factors such as latitude and altitude, by the quality and amount of flora, and by genotype of bee colony (*Costa, Lodesani & Bienefeld, 2012*; *Hatjina et al., 2014*). In a temperate climate, the annual dynamics of the development of a bee colony is as follows: during the winter, a low level of bee population; in the spring, the rapid growth of the colony size; at the beginning of the summer, the peak of the number of bees; further by autumn, a gradual decrease in the bee population (*Hatjina et al., 2014*). In general, our results are consistent with the above pattern, namely, the colony strength in spring was lower than in autumn and much higher in the summer. It should be noted that in Siberia, due to the long (up to seven months) and hard winters with temperatures of −45 °C, the active colony development usually begins in May, and its peak occurs in the middle or end of summer.

We found high correlations between spring and autumn strength of the colony throughout the study period (from 2010 to 2022) and for the period when purebred colonies dominated in the apiary (from 2016 to 2022). In addition, the colony strength in spring (but not in autumn) and the death of bees after wintering (indicator of overwintering ability) were interdependent, and a high correlation of these indicators was shown for the entire period of research (from 2010 to 2022) and for the period when hybrid colonies predominated in the apiary (from 2010 to 2015).

*Infection of bee colonies with parasites and pathogens* may impose limits on colony development (*Hatjina et al., 2014*). Our results support this conclusion: statistically significant correlations were found between the total infestation of bee colonies and their spring and autumn strength, as well as the death of bees. It should be noted that our study was conducted for hybrid bee colonies (from 2010 to 2015), since, starting from 2015, the total infestation of bee colonies has not exceeded 7% (purebred dark forest bees predominated in the apiary).

An assessment of the correlations between individual infections and biological indicators revealed a high correlation only between the *Nosema* infection of bee colonies and the colony strength in spring, as well as the death of bees (Table 2). No correlations were found between the biological traits of colonies and the infection of colonies with varroosis and Chalkbrood disease ($p > 0.05$). On the contrary, a pan-European experiment (a study of bee colonies of different European honey bee genotypes, 2009–2011) showed that the number of adult bees was correlated with *Varroa* infestation level (*Hatjina et al., 2014*).

*The hygienic behavior*. Certain aspects of honey bee colony behavior, such as hygienic behavior are known to influence the *V. destructor* infestation level of a colony (*Meixner et al., 2014*). We evaluated *the hygienic behavior* and *cleanliness of the hive* to characterize the resistance of bee colonies to diseases. High negative correlations are shown between the death of bees after wintering and the cleanliness of the beehive for all test periods.

Another key finding of our study was the difference in biological, behavioral, and economic traits between purebred *A. m. mellifera* (local) and hybrid colonies.

*Local and non-local bee colonies*. Based on the correlations between biological and behavioral traits of colonies, such as the colony strength, the death of bees after wintering, infection with parasites/pathogens, we can state different developmental patterns for

purebred dark forest bees and hybrid colonies. In the dark forest bee, significant correlations were found between autumn and spring colony strength, that is, a weak dependence of colony development in spring on wintering weather conditions. No relationship between the colony strength and other biological indicators was found in the dark forest bee. In hybrids, correlations were found between most of the studied parameters, except for the colony strength in autumn. It can be assumed that the spring development of a hybrid colony significantly depends on wintering conditions: a strong colony in spring leads to rapid development and high egg production of the queen. For the colony development, the significance of both genetic (the origin of bees) and climatic factors, is shown.

To assess the influence of the genetic factor on the main parameters of colonies, we compared the biological, behavioral, and economic traits of purebred (local) and hybrid (non-local) bee colonies in 2012–2014, differing in weather conditions (favorable year 2012 and unfavorable year 2013). Significant differences in biological and behavioral traits between purebred and hybrid colonies were revealed in all years. Significant differences in honey productivity between purebred and hybrid colonies were shown only in the unfavorable year 2013 (Table 4).

Like the results of *Hatjina et al. (2014)*, in our study, colonies of local origin (*A. m. mellifera*) had significantly higher strength than non-local hybrid colonies. Behavioral characteristics also differed: compared to hybrids, dark forest bees were more gentleness and had a low or controlled ability to swarm. It was previously noted that hybridization reduces gentleness of bee colonies (*Uzunov et al., 2014*). We also observed a significantly lower infestation of the local dark forest bees compared to hybrids, which may indicate a specific local adaptation of the *A. m. mellifera* subspecies. For example, a study of European honeybees of different genetic origins (2009–2011) showed that the survival of bee colonies with queens of local origin was statistically significantly higher than colonies with non-local queens (*Büchler et al., 2014*). For *A. m. carnica*, local non-breeding colonies showed high overwintering index, active spring development, resistance to *V. destructor*, and good local adaptation, while breeding strains had better scores for defensive behavior, calmness, and swarming (*Kovačić et al., 2020*).

Of particular interest is the assessment of honey productivity in local and non-local bee colonies. In our study, the honey productivity of local (*A. m. mellifera*) and non-local (hybrid) colonies differed statistically significantly, and dark forest bee colonies were more productive (the average honey productivity was 61–65 kg in 2017–2022) than hybrids (the average honey productivity was 32–44 kg in 2010–2015; only in the favorable year of 2012, honey productivity was high, more than 57 kg). Comparative characteristics of purebred *A. m. mellifera* and hybrid colonies from 2012 to 2014 showed statistically significant differences in honey productivity in the unfavorable year 2013. The results showed a significant dependence of hybrid (non-local) colonies on environmental factors, and in favorable conditions the colonies developed well and were productive, but when weather conditions worsen, the colonies were poorly adapted. Purebred (local) colonies are well adapted to the conditions of Siberia and are highly productive in both favorable and unfavorable years.

It should be noted that in Siberia, both purebred *A. m. mellifera* and hybrid colonies showed high average honey production compared to other studies. For comparison, in Croatia, in *A. m. carnica*, the long-term average of honey production/colony is in the range of 18.2 to 41.4 kg (*Štefanić et al., 2004*). In an unfavorable year 2016, the average amount of extracted honey was very low (10.9 kg) (*Kovačić et al., 2020*). In a pan-European experiment, the overall average honey yield was 23.4 kg (*Hatjina et al., 2014*). Commercial strains of subspecies *A. m. ligustica* and *A. m. carnica* had higher honey yields (about 40 kg) compared to the bee colonies belonging to subspecies *A. m. mellifera* and *A. m. macedonica* (minimum 15.2 kg). Many studies, including our experiment, show a strong influence of environmental conditions on the honey productivity of bee colonies. Additionally, the availability of adequate foraging resources and colony strength affect honey production in the colony (*Woyke, 1984*; *vanEngelsdorp & Meixner, 2010*; *Hatjina et al., 2014*).

In conclusion, *A. m. mellifera* colonies well adapted to the local abiotic environment of Siberia (temperate continental climate) showed significantly better scores for the main commercially recognized biological, behavioral, and economic traits. Dark forest bee colonies were strong, overwintering ability, disease resistance, and highly productive. On the contrary, a low overwintering ability and a slower spring development as well as weak disease resistance, indicate a lack of adaptation of hybrid colonies to environmental factors in Siberia.

Local honeybees are better adapted to local environmental factors such as vegetation, flowering patterns, and climate change, as well as local beekeeping methods. Local bees are constantly exposed to local stressors, including parasites and pathogens, and may have better resources to resist negative factors (*Le Conte et al., 2007*; *Francis et al., 2014*; *Meixner et al., 2014*). On the contrary, non-native bees have fewer resources to survive and are characterized by lower biological and economic traits. Thus, the conservation of local honeybees should be an important priority, as well as the use of local honey bee breeds in breeding programs.

## CONCLUSIONS

In conclusion, our experiment demonstrated that the dark forest bee is more productive, gentleness, and disease resistant compared to hybrids. Our results comparing the vitality and performances of purebred and hybrid bees indicate the need to breed purebred bee colonies in apiaries. The dark forest bee is the most adapted to the conditions of Siberia (long cold wintering, short summer flow and resistance to some diseases). Subspecies *A. m. mellifera* is characterized by high productivity and overwintering ability. At present, in Europe, the subspecies *A. m. mellifera* is recognized as endangered, in connection with which the question of preserving the populations and gene pool of the dark forest bee is of biospheric importance. Siberia represents a unique region for the conservation of the dark forest bee, *A. m. mellifera*, from uncontrolled introgression of imported subspecies.

### Funding
The authors did not receive funding for this work.

### Competing Interests
The authors declare that they have no competing interests.

### Author Contributions
- Nadezhda V. Ostroverkhova conceived and designed the experiments, performed the experiments, analyzed the data, prepared figures and/or tables, authored or reviewed drafts of the article, and approved the final draft.
- Svetlana A. Rosseykina conceived and designed the experiments, performed the experiments, analyzed the data, authored or reviewed drafts of the article, and approved the final draft.
- Ilona A. Yaltonskaya performed the experiments, analyzed the data, prepared figures and/or tables, and approved the final draft.
- Michail S. Filinov performed the experiments, analyzed the data, prepared figures and/or tables, and approved the final draft.

### Data Availability
The raw measurements are available in the Supplemental Files.

### Supplemental Information
Supplemental information for this article can be found online at http://dx.doi.org/10.7717/peerj.17354#supplemental-information.

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
