# Peer review of "Estimates of the vitality and performances of Apis mellifera mellifera and hybrid honey bee colonies in Siberia: a 13-year study"

_PeerJ, doi:10.7717/peerj.17354_

## Round 0.1 · original submission · Major Revisions

The study represents a descriptive approach to efforts in Russian Siberia to maintain and select for local Apis mellifera mellifera populations as a tool to improve beekeeping in this region. The paper is very well written and clearly meets the minimum publishing standards for scientific work in PeerJ’s model . However based on the reviewer's evaluation, the manuscript needs major changes. There are several questions raised by reviewer no. 3 that need to be clarified, especially in the material and methods sections.

Reviewer 1 ·

Basic reporting

See Below

Experimental design

See Below

Validity of the findings

See Below

Additional comments

Review Comments
PeerJ 93299
Summary
This paper represents a descriptive, “data reporting” approach to efforts in Russian Siberia to maintain and select for local Apis mellifera mellifera populations as a tool to improve beekeeping in this region. While the presentation and analysis of this data set is archaic, it is not incorrectly done, and the paper is altogether vert well written. Overall, this manuscript clearly meets the minimum publishing standards for scientific work in PeerJ’s model.
Major Comments
The abstract and introduction are very well written, and I comment the authors for a thorough overview of the literature and extremely effective writing.
The methods are adequate given the length of time this study – many of the “standards” now used in honey bee health research were not available or established at the beginning of this data set, and so the differences in approach used by these authors should be understood in that context.
My main criticism is with the statistics and graphical presentation; while the statistical approach as done is legitimate, it is rudimentary and obscures much of the data. I would strongly encourage the authors to consider repeating their statistics using R and a linear-modelling based approach, as this would help modernize the manuscript and improve the accessibility of the data. Currently, only summary statistics are provided in the supplement.
Correspondingly, the results section is long and very descriptive and detailed: it is difficult to identify what the core results are which drive the narrative of the manuscript or what the main points the reader should be aware of are.
The discussion is mostly descriptive but well written, and the points made are supported by the data provided.
Minor Comments
L31 -“colonies were more survivability, productive, and gentleness” – “colonies showed more gentleness, productivity, and survivorship”
L32- delete “significantly”
L53 –“insufficient vitality” is an unclear phrase in a scientific context, I’d recommend using some more phenotype-explicit language
L54- “locally“ -> ”local”
L128- “research algorithm” is unusual / unclear phrasing, suggest a different subheading

·

Basic reporting

In Introduction: line number 46 de Miranda, Genersch, 2020 in the references De Miranda, Genersch, 2020 (what is the right)
In Materials and Methods: line number 120 Ostroverkhova et al., 2018,2020 but in the References Ostroverkhova, 2020 (what is the right).

Experimental design

Materials and Methods: the paragraph under the title Colony strength or colony size from line 169 to line 175 it need to write a reference.

Validity of the findings

no comment

Reviewer 3 ·

Basic reporting

Overall, the authors use various different terms to describe the same thing, which makes it very difficult to follow. The introduction is missing information on how genetic lineages can be differentiated and what lineages there are in the first place. There are many redundant sentences, and repetitions which seem to be used as space fillers.

Experimental design

I do believe that the research is interesting and valuable, especially given the isolated location of the study area. However, it was not clear how many colonies were assessed during which time period specifically. It was difficult to identify what the actual sample sizes are. The authors mentioned early on that there were 120-200 colonies, but then referred to 64 and finally used the mean across all colonies for comparisons. This is problematic, as I do not know how many colonies were compared, how many of them survived, etc. Additionally, I do not think that some of the measured variables are the best choice. For example, the authors did not provide a reference for the "colony strength" assessment which is based on weight extrapolated from frame coverage with bees. More frequently, scientists refer to the bee population as colony strength. The "overwintering ability" was described as "death of bees" which in the context of honey bee colonies being a superorganism, does not seem to make sense. A colony either survives the winter or not, that is not necessarily dependent on how many individual bees survived. Finally, the identification of diseases and resistance thereof were performed properly, or at least not explained properly.

Validity of the findings

The authors claim that "purebred" bees are better than hybrids. However, these comparisons are made based on the mean across ... something that the authors did not describe, and measures for the two different group were taken during different time periods (i.e., hybrids were kept from 2010-2015, while purebreds were kept thereafter until 2020). Thus, I think the authors should not directly compare hybrids and purebreds. The statistical test that the authors used would not be able to account for between year variability. Overall, I don't think the statistical outputs provided in the results section are sufficient to support those claims.

Annotated reviews are not available for download in order to protect the identity of reviewers who chose to remain anonymous.

---

## Round 0.2 · accepted · Accept

Dear Editors,

Thank you for addressing all of the reviewers' comments and editing the manuscript. I went through the revised version and noticed that more information about the evolutionary lineages of the honey bee Apis mellifera was added. After editing the introduction; materials and methods; results sections and adding more references, I confirm that the manuscript is now ready to be published at the PeerJ journal.

Thank you very much for your time and consideration.

Reviewer 1 ·

Basic reporting

Article is well written, thorough, and suitable for publication.

Experimental design

The work is scientifically sound.

Validity of the findings

The work is scientifically sound.

Additional comments

I look forward to seeing this revised work published.

·

Basic reporting

See below

Experimental design

See below

Validity of the findings

See below

Additional comments

I do not have any observations about this manuscript after the first correction.